# Optimization of Lignin–Cellulose Nanofiber-Filled Thermoplastic Starch Composite Film Production for Potential Application in Food Packaging

**DOI:** 10.3390/molecules27227708

**Published:** 2022-11-09

**Authors:** Tawakaltu AbdulRasheed-Adeleke, Evans Chidi Egwim, Emmanuel Rotimi Sadiku, Stephen Shaibu Ochigbo

**Affiliations:** 1Department of Biochemistry, Federal University of Technology P.M.B. 65, Minna 920001, Niger State, Nigeria; 2Africa Centre of Excellence (ACE) for Mycotoxin and Food Safety, Federal University of Technology, P.M.B. 65, Minna 920001, Niger State, Nigeria; 3Department of Chemical, Metallurgical and Materials Engineering, Polymer Division, Pretoria West Campus 0183, Tshwane University of Technology, Pretoria 0183, South Africa; 4Department of Chemistry, Federal University of Technology, P.M.B. 65, Minna 920001, Niger State, Nigeria

**Keywords:** lignin, nanocomposite, cellulose nanofiber, optimization, percent elongation, starch, tensile strength, Young’s modulus

## Abstract

The optimization of the production of thermoplastic starch (TPS) bionanocomposite films for their potential application in food packaging was carried out, according to the Box–Wilson Central Composite Design (CCD) with one center point, using Response Surface Methodology (RSM) and fillers based on lignin and nanofiber, which were derived from bamboo plant. The effects of the fillers on the moisture absorption (*MAB*), tensile strength (*TS*), percent elongation (PE) and Young’s modulus (*YM*) of the produced films were statistically examined. The obtained results showed that the nanocomposite films were best fitted by a quadratic regression model with a high coefficient of determination (*R*^2^) value. The film identified to be optimum has a desirability of 76.80%, which is close to the objective function, and contained 4.81 wt. % lignin and 5.00 wt. % nanofiber. The *MAB*, *TS*, *YM* and *PE* of the identified film were 17.80%, 21.51 MPa, 25.76 MPa and 48.81%, respectively. The addition of lignin and cellulose nanofiber to starch composite was found to have reduced the moisture-absorption tendency significantly and increased the mechanical properties of the films due to the good filler/matrix interfacial adhesion. Overall, the results suggested that the produced films would be suitable for application as packaging materials for food preservation.

## 1. Introduction

Synthetic plastics have dominated every field of human activity, particularly the packaging industries [1]. Despite their many merits, synthetic plastics have been a major environmental concern for some time. Since they are non-biodegradable and also dependent upon a non-renewable petroleum resource, the blooming usage of these plastics has caused grave energy crises, as well as environmental pollution associated with their disposal, including damage to the eco-system, water supplies and sewer systems, as well as rivers and streams. As a result, great attention is being drawn to natural polymers, e.g., starch, due to its potential as a resource for making environmentally friendly useable products to replace those derived from petroleum. The reason for starch being favored for this purpose is because it is inherently biodegradable, renewable, cost effective and available from many plants. In order to make it suitable for industrial application, such as packaging, starch is usually processed in the presence of a plasticizer to obtain a product called thermoplastic (TPS). However, TPSs alone cannot be employed for packaging because they have low mechanical strength and high water sensitivity [2], which falls below the requirements of a packaging material. Thus, in order to overcome these drawbacks, there is the need for the incorporation of reinforcing fillers, such as cellulosic fibers, whiskers, nanofibers, etc., to produce new and inexpensive starch biocomposites with improved properties [3,4,5,6,7,8]. Lignin and nanofibers are potential fillers which are currently in great demand due to their natural abundance and susceptibility to biodegradability, being derived from plants. The term “lignin”, from the Latin word *lignum*, meaning “wood”, was first used by Swiss botanist Candolle [9]. Lignin, a by-product that is mainly obtained from pulp and the paper industry, is the largest aromatic polymer in nature. Although its exact molecular structure is still subject to controversy, lignin is believed to result from the dehydrogenative polymerization of three monomer species, namely p-coumaryl alcohol, coniferyl alcohol, and sinapyl alcohol. The amounts and proportions of the main functional reactive groups such as hydroxyl, methoxyl, carboxyl and carbonyl groups in lignin vary according to the plant species and extraction processes applied. Together with cellulose and hemicellulose, lignin forms the structural components of trees and various plants, and constitutes the most common natural polymers from plants. When lignin and cellulosic fibers are used as fillers in natural polymers, the resulting products are eco-friendly and have improved physical properties. Organic membranes prepared from cellulose derivatives usually have low mechanical strength as well as poor resistance to oxidation. In order to overcome these limitations, cellulose-based films make use of lignin [10]. The use of lignin and/or cellulosic fibers to improve the properties of TPS intended to be processed as packaging materials has been reported by a number of researchers. For example, in a study carried out by Kaushik et al. [11], cellulose nanofibrils were extracted from wheat straw using steam explosion, acidic treatment and high-shear mechanical treatment. These nanofibrils were dispersed in thermoplastic starch (TPS) using a Fluko high-shear mixer in varying proportions, and films were casted out of these nanocomposites. The results showed that the mechanical properties increased with the nanofiber concentration. The barrier properties also improved with the addition of nanofillers up to 10%, but further addition deteriorated the properties due to possible fiber agglomeration. In order to improve the mechanical properties and the resistance to water absorption of thermoplastic starch (TPS), Kaewtatip and Thongmee [12] employed kraft lignin (KL) and esterified lignin (EL) as fillers for the TPS matrix. EL was produced via esterification of the KL hydroxyl groups. The TPS/lignin composites were prepared using compression molding. The amount of each of the lignins used in the composites was 5 wt. % (dry starch basis). The TPS and composites were investigated using Fourier-transform infrared spectroscopy (FTIR), water absorption and tensile testing. The FTIR spectra showed that the interaction between the TPS and each lignin caused the peak of the OH stretching shift to lower the wavenumber. This result indicated that both the TPS/KL and the TPS/EL composites had improved mechanical properties over TPS. The tensile strength of the TPS/KL and TPS/EL composites was higher than for the TPS by about 17% and 32%, respectively. In addition, the presence of lignins in the TPS matrix significantly decreased the water-absorption properties. The combination of lignin and cellulosic fiber in the production of a new TPS composite was created by Narchamnan and Sakdaronnarong [13], using a laccasse-mediator system to enhance the binding efficiency of natural fibers and lignin compounds into a cassava starch composite matrix. In this work, violuric acid (VA) was tested for its effect as a mediator for laccase treatment. The influence of different fiber, lignin and water contents of the biocomposite was statistically investigated. The results showed that adding 15% (*w*/*w*) fibers into biocomposite at 44% (*w*/*w*) water content increased the flexural strength and modulus by four times compared with the control. A combination of fibers + VA gave the greatest enhancement of the modulus at 1140% and flexural strength at 375.8%, as much as neat starch biocomposite. The presence of fibers, lignin and VA as mediators for laccase treatment substantially enhanced the water resistance of starch biocomposite, which was detected by a change in the water drop contact angle on the biocomposite surface.

So far, studies based on the simultaneous incorporation of lignin and cellulose nanofibers into thermoplastic starch composite film, besides the one referenced above, have not been reported in the literature. Moreover, by changing the sources of the lignin and fibers used in TPS, new biocomposites with unique properties can result. In this work, the lignin and nanofibers were obtained from bamboo plant (*Bambusa vulgaris*). Further to this, work on the optimization of the experimental variables for the development of flexible films from starch for food packaging is limited, hence, the justification for the present study. Therefore, the objective of this research is to produce a lignin–cellulose nanofiber-filled thermoplastic starch composite film, with the optimization of key process variables, for potential application in food packaging.

## 2. Results and Discussion

### 2.1. Film Morphologies

The SEM micrographs of the pure starch (representing the negative control sample) and that of lignin–cellulose nanofiber-filled thermoplastic starch composite film (the target sample) are shown in Figure 1 and Figure 2, respectively. The SEM micrograph of the pure starch, as seen (Figure 1), showed the characteristic near-spherical morphology of the cassava starch granules with varying sizes [14]. The micrograph showed that the starch granules were generally in the order of sizes that ranged between 2 and 7 µm. This morphology is, however, destructured during the starch gelatinization process, subsequent to the incorporation of the fillers, in order to pave the way for good mixing in the composite formation stage. In Figure 2, the SEM micrograph shows that the fillers (lignin and cellulose nanofiber) were homogeneously distributed within the matrix of the starch nanocomposite film, a situation that usually results in improved mechanical properties. The white dots with varying sizes on the composite are considered as the fillers embedded in the starch matrix. The sizes of the cellulose nanofibers and lignin were evaluated using transmission electron microscopy (TEM), and their respective micrographs were as shown in Figure 3a,b. It was observed that the sizes of the cellulose nanofibers varied from 20 to 100 nm, while those of lignin fell between 20 and 200 nm. The determined characteristics of the used cellulose nanofiber are hereby shown in Table 1.

### 2.2. Development of Regression Model Equations by Central Composite Design (CCD) for Lignin–Cellulose Nanofiber-Filled Thermoplastic Starch Composite Film

The quadratic models, in terms of the actual factors used in achieving the desired optimum or ideal films, were obtained from an analysis of variance (ANOVA) (Table 2, Table 3, Table 4 and Table 5), and are given in Equations (1)–(4). The positive signs in the models indicate synergetic effects, whereas the negative signs, antagonistic effects. The model for predicting the moisture-absorption capacity of the lignin–cellulose nanofiber-filled thermoplastic starch composite film in terms of actual factors is given in Equation (1).
(1)MAB=35.53+0.837l−0.321n−0.755ln
$$\langle R^{2}=91.02\%|R_{adj}{}^{2}=85.63\%|R_{pred}{}^{2}=64.30\%\rangle$$
where: *MAB* = moisture-absorption capacity, *l* = lignin (wt. %), *n* = nanofiber (wt. %).

The model was significant at a *p*-value of 0.0048, and the coefficient of determination (*R*^2^) of 0.9102 was obtained. This indicates that the statistical model could explain 91.02% of the variability, while the remaining 8.98 wt. % could not be accounted for by the independent variables [15]. The predicted $R^{2}$ of 64.30% reasonably agrees with the adjusted $R^{2}$ (85.63%), as should normally be the case for model adequacy due to the very small blocking effect of the data used. The precision of the model is high since the adequate precision value obtained was ~11.156 [16].

The model for predicting the percentage elongation of the lignin–cellulose nanofiber-filled thermoplastic starch composite film in terms of actual factors is given in Equation (2).
(2)$$PE=50.35+0.10l-1.09n+0.077ln-0.067l^{2}+0.124n^{2}$$
$$\langle R^{2}=84.49\%|R_{adj}{}^{2}=58.65\%|R_{pred}{}^{2}=-82.39\%\rangle$$
where: *PE* = elongation (wt. %); *l* and *n* retained their usual meaning.

The model was significant at a *p*-value of 0.1792, and the coefficient of determination (*R*^2^) of 0.8449 was obtained. This indicates that the model could describe 84.49 wt. % variability, while 15.51 wt. % was inexplicable by the independent variables [15]. In addition, the negative predicted $R^{2}$ value of −82.39 wt. % implied that the fitted model, though precise (with adequate precision value of 5.37), is not a suitable predictor of the percentage elongation of the starch composite film.

The model for predicting the tensile strength of the lignin–cellulose nanofiber-filled thermoplastic starch composite film with respect to the actual factors is given in Equation (3).
(3)TS=17.32+0.37l+0.48n
$$\langle R^{2}=83.46\%|R_{adj}{}^{2}=77.94\%|R_{pred}{}^{2}=61.72\%\rangle$$
where: *TS* = tensile strength (MPa); *l* and *n* retained their usual meaning.

The model was significant at a *p*-value of 0.0045, and the coefficient of determination (*R*^2^) of 0.8346 was obtained. This indicates that the statistical model could explain the 83.46% value of the variability, while a 16.54% value was inexplicable by the independent variables [15]. In addition, the predicted $R^{2}$ of 61.72 wt. % agrees with the $R^{2}$ (adjusted) of 77.94%, as should normally be the case for adequate model. The model is precise in its capacity to predict the tensile strength of the starch mixture film, since an adequate precision value of 10.91 was recorded [16].

The model for predicting the Young’s modulus of the lignin–cellulose nanofiber-filled thermoplastic starch composite film in terms of actual factors is given in Equation (4).
(4)YM=24.5+0.09l+0.165n
$$\langle R^{2}=95.01\%|R_{adj}{}^{2}=93.34\%|R_{pred}{}^{2}=91.81\%\rangle$$
where: *YM* = Young’s modulus (MPa); *l* and *n* retained their usual meaning.

The model was found to be significant at a 0.0001 *p*-value, and the coefficient of determination (*R*^2^) of 0.9501 was obtained. This indicates that the model could explain 95.01% of the variability, while 4.99% was inexplicable by the independent variables [15]. The predicted $R^{2}$ value of 91.81% reasonably agrees with the $R^{2}$ adjusted value of 93.34% for the adequate model, and this was caused by the small block effect of the data used in generating the model. In addition, the model is precise in its capability to forecast the lignin–cellulose nanofiber-filled thermoplastic starch composite film’s Young’s modulus, since the adequate precision recorded value was 20.51 [16].

### 2.3. Analyses of Response Surfaces

The 3D response surface plots of the joint effects of the independent variables (lignin and cellulose nanofiber) on the *MAB*, *TS*, *YM* and *PE* of lignin–cellulose nanofiber-filled thermoplastic starch composite film, as performed according to the working conditions stated in Table 6, are presented in Figure 4. The 3D response surfaces were plotted in order to investigate the probable interactions among the variables, and to determine the optimum conditions of each factor for the minimum moisture absorption and maximum strength of the starch nanocomposite films.

The response surface plots for the influence of lignin and cellulose nanofiber contents on the *MAB*, *TS*, *YM* and *PE* of the lignin–cellulose nanofiber-filled thermoplastic starch composite film are presented in Figure 4.

The results revealed that the inclusion of cellulose nanofiber content had the most significant effect on the moisture absorption and mechanical properties of the lignin–cellulose nanofiber-filled thermoplastic starch composite film, followed by lignin inclusion content. The *MAB* reduced rapidly as the cellulose nanofiber content increased when compared to that of lignin content. The lignin and cellulose nanofiber contents interacted negatively. This shows that the moisture absorption reduced as the lignin and cellulose nanofiber contents increased. The decrease in *MAB* with the increase in the lignin and cellulose nanofiber contents was because of the strong filler–matrix interaction, which reduced the molecular mobility and diffusivity in the matrix material, thus, limiting the degree of moisture uptake [17]. This result is in accordance with the findings of Khan et al. [18], Mitchelle [19], and Taghizedeh and Sabouri [20], who reported low moisture-absorption rates for cocoyam, tapioca and modified corn starch films, respectively, with an increase in filler loading.

The results also showed that the *TS* and *YM* of the starch nanocomposite film increased as the lignin and cellulose nanofiber contents increased, while *PE* decreased with the increase in lignin and cellulose nanofiber contents. The significant increase in the *TS* and *YM* of the lignin–cellulose nanofiber-filled thermoplastic starch composite film with the increase in filler contents, particularly nanofiber, was due to the small sizes (nano-range), smooth surface and large surface area of these fibers, which produced a good fiber/matrix interaction, thus, improving the composite film’s strength [21]. It might also be that lignin improved the compatibility between the starch and cellulose nanofiber [22]. Moreover, Suarez et al. [23] also reported the fact that a good interfacial region increases stress transfer efficiency from the matrix to the fillers, thereby increasing the composite’s strength.

This result corroborated with that of Wang et al. [22], who confirmed a rise in the *TS* and *YM*, but a decrease in the elongation-at-break of a PLA nanocomposite reinforced with lignin–cellulose nanofiber (L-CNF), as the L-CNF content rose from 25 to 35 wt. %. Patpen et al. [24] also reported the effect of cellulose addition on the *TS* of polylactic acid biocomposite; they observed an increase in this parameter as cellulose loading increased. Tawakkal et al. [25] also reported how kenaf-derived cellulose (KDC)-loaded polylactic affected the material’s tensile properties and documented an improvement in both *TS* and tensile modulus, as KDC loading increased from 30 to 60 wt. %. On the contrary, Sawpan et al. [26] studied the mechanical properties of hemp fiber-reinforced PLA biocomposites and established a non-linear relationship between *TS* and fiber content, indicating that the strength of the biocomposite somewhat decreased as the fiber content increased from 0 to 40 wt. %.

### 2.4. Optimization of Lignin–Cellulose Nanofiber-Filled Thermoplastic Starch Composite Film Production

The result of the optimization of the experimental variables (lignin and cellulose nanofiber contents), showing the desirability function, with respect to the films prepared, is shown in Figure 5. The goal for both the water barrier and mechanical optimization was to minimize the moisture absorption and improve the mechanical property; thus, the target value of the responses, as obtained from the obtained experimental results, was at the lowest and uppermost values, respectively. The result from the optimization showed that increases in the lignin and cellulose nanofiber contents, from 1 to 5 wt. %, significantly affected the desirability of the films. Therefore, it is obvious from the optimization result that the selected film with a desirability closer to the goal is the one that was produced with the blend of 4.81 wt. % lignin and 5.00 wt. % nanofiber at 76.80% desirability. The corresponding *MAB*, *TS*, *YM* and PE of the selected film, apparently the optimum sample, are 17.80%, 21.51 MPa, 25.76 MPa and 48.81%, respectively. This finding is similar to that of Wang et al. [22], who documented optimum values of 21.6 MPa and 21.6% for the *TS* and elongation, respectively, for PLA composites reinforced with L-CNF. Akbar et al. [27], however, obtained optimum values of 50 MPa for the *TS*, 2.15 MPa for the *YM* and 141.07% for elongation, during the optimization of polyvinyl alcohol nanocomposite films. Patpen et al. [24] also reported an optimum value of 46.207 MPa for the optimization of PLA-based biocomposites that were reinforced with cellulose obtained from durian peel.

This study thus affirms that lignin and cellulose nanofiber addition to starch composite minimized *MAB* and improved the mechanical properties of the produced films because of the good filler/matrix interfacial adhesion, as evident by the SEM micrograph. Hence, the produced films can be applied to package food.

## 3. Materials and Methods

### 3.1. Plant Materials

Cassava (*Manihot esculenta* crantz) was purchased from a farm in Minna, Nigeria, whereas bamboo (*Bambusa vulgaris* schrad) was collected from the river banks of Gurara in Izom, Niger State. They were identified and authenticated by a botanist at the National Institute for Pharmaceutical Research and Development (NIPRD) Idu, Abuja. Specimen voucher numbers (NIPRD/H/6792 and NIPRD/H/6793), respectively, were placed at their herbarium for references in the future.

### 3.2. Chemicals

Analytical grade chemicals were employed for the study and included ethanol (BDH chemicals, London, UK); acetic anhydride (Sigma Aldrich, Burlington, MA); sodium hydroxide (Kermel, China); citric acid (BDH chemicals, UK); hydrochloric acid (Griffin and George, UK); sulfuric acid (BDH chemicals, UK); hydrogen peroxide (BDH chemicals, UK); sodium sulphate (BDH chemicals, UK); and ammonium hydroxide (Griffin and George, London, UK). Glycerol (BDH chemicals, UK) was used as the plasticizer for the study.

### 3.3. Preparation Lignin and Cellulose Nanofiber from Bamboo (Bambusa vulgaris Schrad)

Lignin was extracted from bamboo according to standard procedure, as reported by Yong et al. [28], Alemdar and Sain [29] and Ming-Fei et al. [30]. The bamboo stalks obtained were first sun-dried and then chipped into small pieces. The sun-dried pieces of bamboo were ground and screened to obtain a 40–60 µm mesh fraction. The ground bamboo stalk (20 g) was first soaked in NaOH (4% *w*/*w*) at room temperature for 24 h. It was then filtered and washed with distilled water (1 L) until it was free of alkali. The residue was re-dispersed in 1 L of distilled water, filtered again and treated with 10% (*w*/*w*) NaOH at 121 °C in an autoclave for 4 h. Furthermore, the residue was again washed in distilled water to free it of residual alkali (1 L) and filtered. Lignin was precipitated from the filtrate by acidifying it to pH 2 with H_2_SO_4_. The precipitates were separated from the mixture by filtration. The separated lignin was washed with water several times and then oven-dried at 40 °C. In order to obtain bamboo fiber, the supernatant liquid left after the alkali treatment was bleached in 8% (*v*/*v*) H_2_O_2_ at room temperature for 24 h. Finally, the material was again washed and filtered as before to obtain bamboo fiber. The bamboo fiber was then converted to nanofiber using acid hydrolysis [31]. The bamboo fibers obtained after lignin removal were steeped in HCl (10% *w*/*w*) with ultrasonic agitation at 60 °C for 2 h using an Ultrasonicator (SB25-12DT, Scientz, Ningbo, China). The material was given a final wash and then placed in a high shear homogenizer (Heidolph DIAX 900, Burladingen, Germany for 15 min to produce bamboo nanofibers.

### 3.4. Experimental Design and Optimization of Starch Nanocomposite Film Production

Design Expert software (version 7.0, Start-Ease Inc., Minneapolis, MN, USA) was employed for the experimental design, while Response Surface Methodology (RSM) was used for optimizing the conditions required for preparing the starch nanocomposite films. As a result, 1 center-point Box–Wilson Central Composite Design (CCD) was utilized. After the designed experiment was performed, linear regression was used to obtain the results. The design consisted of 9 experimental runs. The RSM considered the effect of two variables: lignin content (wt. %) and cellulose nanofiber content (wt. %) used as fillers in the nanocomposite preparation with 5 levels each. The response functions measured were moisture absorption (*MAB*), tensile strength (*TS*), percent elongation (*PE*) and Young’s modulus (*YM*). An analysis of variance (ANOVA) was employed to analyze the obtained data, in order to determine the interactions that exist between the process variables and the responses. Accurate and proper models were picked at *p* < 0.05 and had a significantcorrelation. The fitting model’s quality was expressed by the coefficient of determination *R*^2^ and adjusted *R*^2^. The factors’ level with their codings are shown in Table 7.

In the optimization selection, two factors (lignin and cellulose nanofiber contents) were considered in order to build desirability indices. The objective was to reduce the *MAB* while improving *TS*, *PE* and *YM*; therefore, the target value of the responses was lowest for *MAB* and highest for *TS*, *PE* and *YM* from the experimental results obtained.

### 3.5. Preparation of Lignin–Cellulose Nanofiber-Filled Thermoplastic Starch Composite Film

Cassava starch (2 g) granules were dispersed in 50 mL of distilled water and heat was applied at 70 °C for 20 min under constant stirring over a magnetic stirrer. Glycerol (50 wt. % based on dry cassava starch content) was added to the dispersion while the heating at 70 °C was continued under constant stirring speed for the next 2 min. Next, the lignin and cellulose nanofiber (varied wt. %, with respect to the dry cassava starch content, and based on the statistical formulation of the Central Composite Design adopted) were added to the dispersion under the same conditions for another 2 min. Before being introduced into the plasticized starch mixture, the cellulose nanofibers were sonicated for 10 min by using a 60 W rated Sonicator. The mixture of dispersion was then cast into a mold and oven-dried at 50 °C using a still-air oven for 18 h, in order to obtain dry lignin–cellulose nanofiber-filled thermoplastic starch composite films [32,33], whose average thickness was found to be 0.12 mm. The control (TPS) sample was also prepared using the same process mentioned above, except that there were no fillers added to it. All films were conditioned at 55 ± 5% RH and 25 ± 2 °C before testing their permeability and mechanical properties, as described by the ASTM standard D882-09 [34] and Detduangchan et al. [35]. The conditioning was performed by inserting the films into desiccators containing a saturated solution of Mg (NO_3_)_2_•6H_2_O for 72 h. Part of the films were tested for water absorption, *TS*, *PE* and *YM*, and the rest, which were for other tests, were kept in plastic bags and inserted into desiccators.

### 3.6. Characterization of Lignin–Cellulose Nanofiber-Filled Thermoplastic Starch Composite Film Water-Absorption Test

The film pieces (20 mm × 20 mm) were pre-conditioned by drying in the oven at 50 °C for 24 h and then weighed to determine the dry weight. They were then immersed in a bath containing distilled water at room temperature. The film samples were removed from distilled water after intervals of 1, 3, 5, 7, 9 and 11 h and, after wiping off the excess water on their surfaces with tissue, their weights were determined. The water-absorption capability (WAC) was, thus, calculated using Equation (5) [32,36]:(5)$$\mathrm{WAC}\;\mathrm{wt}.\;\%=\frac{\left(\mathrm{Wwet}\;-\mathrm{Wdry}\right)}{\mathrm{Wdry}}\times 100\;$$
where: Wwet = Wet specimen weight and Wdry represents the dry specimen weight.

### 3.7. Mechanical Properties

The *TS*, *PE* and the *YM* values were determined with a universal testing machine (DBBMTCL-2500kg Testometric, Rochdale, UK), according to the ASTM D882 Standard [34]. Prior to testing, each sample was conditioned at a temperature of 25 °C and a relative humidity (RH) of 55% for 24 h. The average thickness of the samples was about 0.12 mm. The tensile test was carried out using 1.3 mm/min crosshead speed. Each of the determinations was obtained from triplicate specimens.

### 3.8. Scanning Electron Microscopy (SEM)

The morphological structures of the lignin–cellulose nanofiber-filled thermoplastic starch composite films were determined using SEM (Zeiss Auriga HRSEM) at an accelerating voltage of 15 kV. The as-prepared samples were, respectively, placed on a stub with a double-sided adhesive tape, after which they were coated with a thin layer of gold. The micrographs were captured using a magnification of 350 times the original specimen size [33,37].

### 3.9. Statistical Analysis

Design Expert Software (version 7.0) was employed to analyze the obtained data. The determination of interaction effects between the factors and a quadratic surface plot was generated using an analysis of variance (ANOVA). The model adequacy was examined using the ANOVA, a normal probability plot and a residual plot, according to the method described in the literature [38]. An F-Test was also employed to determine the model’s statistical significance and the regression coefficients’ significance.

## 4. Conclusions

In conclusion, the film with a desirability of ~76.80%—which is closest to the objective function—and containing 4.81 wt. % lignin and 5.00 wt. % cellulose nanofiber, was selected as the optimum sample. The *MAB*, *TS*, *YM* and *PE* of the selected film were 17.80%, 21.51 MPa, 25.76 MPa and 48.81%, respectively. This film presented the maximum mechanical potency and minimal moisture-absorption capacity. The addition of lignin and cellulose nanofiber concurrent to the TPS matrix evidently caused a decrease in moisture absorption, while at the same time an improvement in the mechanical properties of the films. Consequently, the prepared films have the potential to be employed as films for packaging foods.

## Figures and Tables

**Figure 1 molecules-27-07708-f001:**
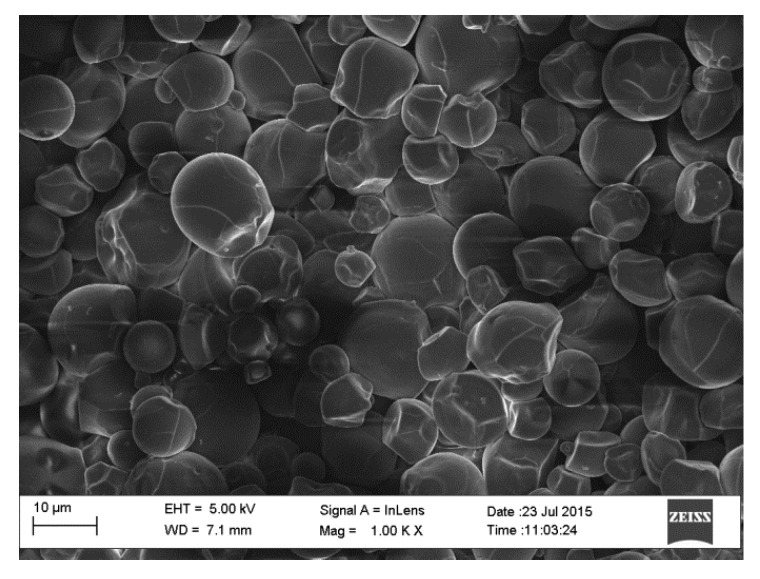
SEM micrograph of pure cassava starch film.

**Figure 2 molecules-27-07708-f002:**
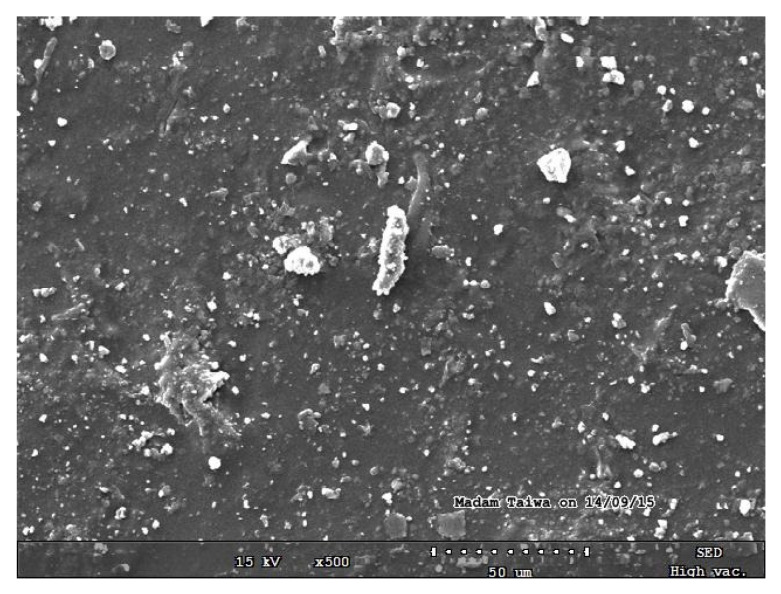
SEM micrograph of lignin–cellulose nanofiber-filled thermoplastic starch composite film.

**Figure 3 molecules-27-07708-f003:**
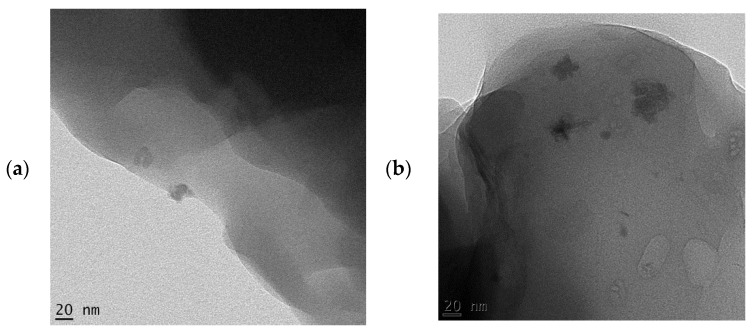
TEM micrographs of (**a**) cellulose nanofiber; (**b**) lignin fillers.

**Figure 4 molecules-27-07708-f004:**
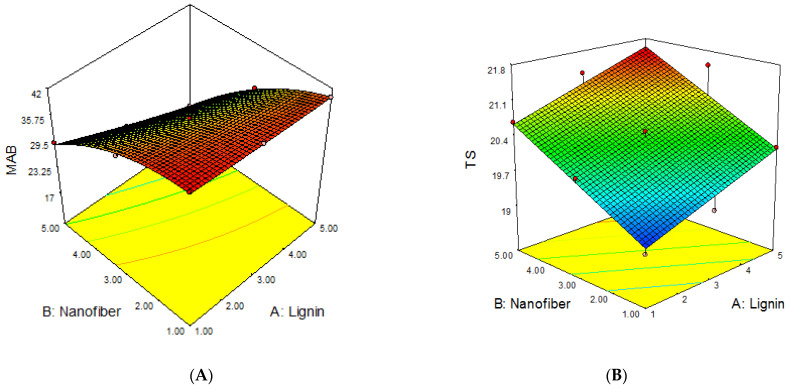
The influence of filler contents on (**A**) *MAB*, (**B**) *TS*, (**C**) *YM* and (**D**) *PE* of the lignin–cellulose nanofiber-filled thermoplastic starch composite film.

**Figure 5 molecules-27-07708-f005:**
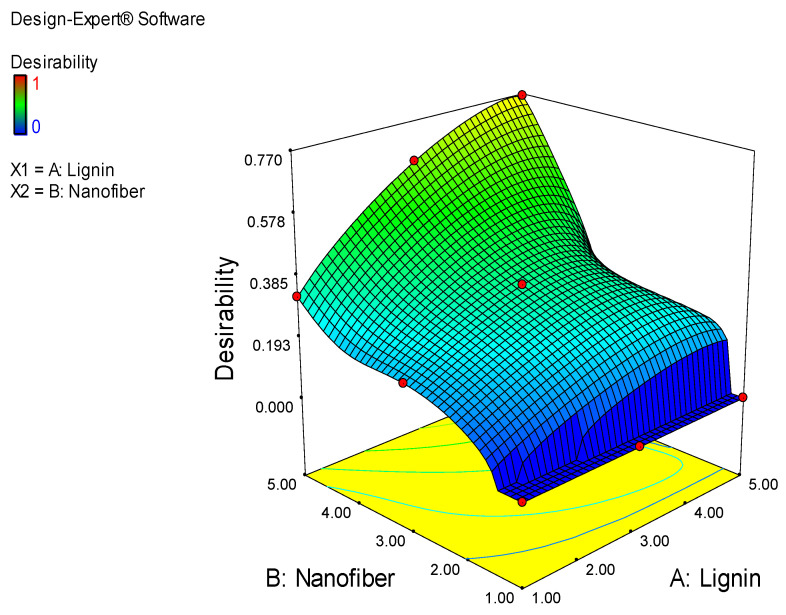
Desirability function of selecting the ideal film prepared.

**Table 1 molecules-27-07708-t001:** The characteristics of the used cellulose nanofiber prepared from bamboo (*Bambusa vulgaris* schrad).

TGA Data			
T_onset_/°C	Peak Temperature/°C	End Degradation Temperature/°C	Residue/wt. %
200.05	328.00	500.58	12.20
**Size range/nm**			
TEM and particle size distribution	20–100		

**Table 2 molecules-27-07708-t002:** ANOVA for response surface quadratic model for *MAB* of lignin–cellulose nanofiber-filled thermoplastic starch composite film.

Source	Sum of Squares	df	Mean Square	F Value	*p*-Value Prob > F	
Model	245.9	3	81.97	16.89	0.0048	Significant
A-lignin	48.91	1	48.91	10.08	0.0247	
B-cellulose nanofiber	160.52	1	160.52	33.08	0.0022	
AB	36.47	1	36.47	7.52	0.0407	
Residual	24.26	5	4.85			
Cor total	270.17	8				
Std. dev.	2.2		R-squared	0.9102		
Mean	28.28		Adj R-squared	0.8563		
C.V. wt. %	7.79		Pred R-squared	0.643		
PRESS	96.44		Adeq precision	11.156		

**Table 3 molecules-27-07708-t003:** ANOVA for response surface quadratic model for TS of lignin–cellulose nanofiber-filled thermoplastic starch composite film.

Source	Sum of Squares	df	Mean Square	F Value	*p*-Value Prob > F	
Model	8.86	2	4.43	15.14	0.0045	Significant
A-lignin	3.27	1	3.27	11.18	0.0155	
B-cellulose nanofiber	5.59	1	5.59	19.09	0.0047	
Residual	1.76	6	0.29			
Cor total	10.61	8				
Std. dev.	0.54		R-squared	0.8346		
Mean	19.88		Adj R-squared	0.7794		
C.V. wt. %	2.72		Pred R-squared	0.6172		
PRESS	4.06		Adeq precision	10.908		

**Table 4 molecules-27-07708-t004:** ANOVA for response surface quadratic model for YM lignin–cellulose nanofiber-filled thermoplastic starch composite film.

Source	Sum of Squares	df	Mean Square	F Value	*p*-Value Prob > F	
Model	0.85	2	0.42	57.09	0.0001	Significant
A-lignin	0.2	1	0.2	26.37	0.0021	
B-cellulose nanofiber	0.65	1	0.65	87.81	<0.0001	
Residual	0.045	6	0.00744			
Cor total	0.89	8				
Std. dev.	0.086		R-squared	0.9501		
Mean	25.26		Adj R-squared	0.9334		
C.V. wt. %	0.34		Pred R-squared	0.9181		
PRESS	0.073		Adeq precision	20.514		

**Table 5 molecules-27-07708-t005:** ANOVA for response surface quadratic model for PE of lignin–cellulose nanofiber-filled thermoplastic starch composite film.

Source	Sum of Squares	df	Mean Square	F Value	*p*-Value Prob > F	
Model	1.43	5	0.29	3.27	0.1792	Not significant
A-lignin	0.11	1	0.11	1.22	0.3502	
B-cellulose nanofiber	0.32	1	0.32	3.6	0.154	
AB	0.38	1	0.38	4.29	0.1302	
A^2^	0.14	1	0.14	1.62	0.2922	
B^2^	0.49	1	0.49	5.62	0.0985	
Residual	0.26	3	0.088			
Cor total	1.69	8				
Std. dev.	0.3		R-squared	0.8449		
Mean	48.76		Adj R-squared	0.5865		
C.V. wt. %	0.61		Pred R-squared	−0.8239		
PRESS	3.09		Adeq precision	5.373		

**Table 6 molecules-27-07708-t006:** RSM factorial design matrix for lignin–cellulose nanofiber-filled thermoplastic starch composite film.

Run	Factors		Responses			
	Lignin (wt. %)	Cellulose Nanofiber (wt. %)	*MAB* (%)	*TS* (MPa)	*YM*	*PE* (%)
1	3.0	1.0	32.558	18.370	24.960	49.300
2	1.0	3.0	31.818	19.250	25.050	48.250
3	1.0	1.0	32.787	18.070	24.800	49.650
4	5.0	3.0	28.571	21.510	25.520	48.525
5	3.0	5.0	22.034	20.510	25.675	49.000
6	5.0	5.0	15.590	20.980	25.740	48.575
7	5.0	1.0	31.884	19.750	25.100	48.500
8	1.0	5.0	28.571	20.490	25.425	48.500
9	3.0	3.0	30.693	19.970	25.100	48.500

**Table 7 molecules-27-07708-t007:** Experimental variables and their coded levels of variables levels for CCD.

Variables	Units	Coded Levels		
		−1	0	+1
Lignin	wt. %	1.0	3.0	5.0
Cellulose nanofiber	wt. %	1.0	3.0	5.0

## Data Availability

Not applicable.

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
