# Peer review of "Optimization of Lignin–Cellulose Nanofiber-Filled Thermoplastic Starch Composite Film Production for Potential Application in Food Packaging"

_molecules, 2022, doi:10.3390/molecules27227708_

Round 1

Reviewer 1 Report

The current format cannot be accepted. The specific reasons are as follows

1.     The introduction is poor. The authors should describe the research background and current situation more.

2.     The second chapter should be the experimental methods and details.

3.     Figure 1 is not clear. The size bar in Figure 1 should be clearly displayed.

4.     The innovation points in the article need to be highlighted.

5.     There are some grammatical errors in English.

6.     The abstract and conclusions need to be rewritten. Here, the authors should focus on describing the main findings of this study.

Reviewer 2 Report

The author uses lignin and nanofibers to improve the properties of TPS, and believes that lignin can improve the interfacial compatibility between nanofibers and substrate. The premise of publishing this work is that the author can solve the following problems:

1. The microstructure image of modified starch films was observed by SEM. It was considered that the uniform distribution of fillers was helpful to improve the properties of the films. The authors should add the SEM images of films with different filler addition ratios to conduct more detailed theoretical analysis, and whether the sea islands structure is suitable for explaining this phenomenon.

2. The format of tables 2 and 4 needs to be modified.

3. In Table 2, the difference between Adj R-Squared and Pred R-Squared is greater than 0.2. Why did author think they are reasonable?

4. The author takes TPS without filler as the negative control group, and I suggest that the performance data of the negative control group be listed separately. In addition, the author should test the film with only nanofibers, which will have a better effect.

5. The test data of nanofibers are not given, such as length, aspect ratio, etc. In addition, whether nanofibers of different sizes can support the author's conclusion.

Round 2

Reviewer 1 Report

The author have made a large number of modifications according to my suggestions, so it can be accepted in present form.

Author Response

Thank you for your invaluable constructive review comments and your satisfaction with modifications made according to the observed comments. Please, note that I have again reviewed the English language style and, to the best of my knowledge, haven't overlooked any grammatical error. Once again, thank you for adding value to the paper by your review work. 

Reviewer 2 Report

I think the author needs to seriously consider and answer the following two questions. The article is acceptable for publication only if the author gives reasonable explanations.

1. I think it is impossible to prove the effect of the presence of individual fillers on film performance if there is no test data for the negative control group and nanofibers/lignin group. The author should seriously consider and respond to this problem.

2. The author should carefully check the format of Table 2 to Table 5, as Table 2 and the first row of Table 4 are separated into two parts, while Table 3 and Table 5 are not divided. This format error must be corrected.

Author Response

I am very much impressed by your high profile review work and your comments well taken in good faith. I have addressed the two issues you identified that must be explained;

1. We have provided the scanning electron microscopy (SEM) micrograph for the pure cassava starch, being the negative control sample, as required in order to provide the needed contrast against the final composite film. Furthermore, the TEM micrographs of each of both the nanofiber and lignin which were the nanofillers incorporated into the starch matrix to indicate their respective size ranges.

Round 3

Reviewer 2 Report

The author has supplemented the introduction, added the SEM image of the pure starch film as a negative control and the TEM image of the modified film, corrected the formatting errors in the tables, and I think this article is acceptable and published.